# Identifying predictors of formal help-seeking for premenstrual symptoms: A machine learning analysis of symptom, functional impairment and barriers data

Erin L. Funnell🆔, Nayra A. Martin-Key, Jakub Tomasik, Sabine Bahn*

Cambridge Centre for Neuropsychiatric Research (CCNR), Department of Chemical Engineering and Biotechnology, University of Cambridge, Cambridge, United Kingdom

* sb209@cam.ac.uk

## Abstract

Despite the potential severity and burden of premenstrual symptoms, few appear to seek formal care. Given that access to many therapeutic interventions requires formal help-seeking, it is important to understand predictors of this health behaviour. This study employed machine learning to identify symptoms, functional impairment, and barriers to accessing care that predict formal help-seeking for premenstrual symptoms. Data was collected from a UK-based sample using online survey software and explored using descriptive analysis. Group differences in ordinal and categorical data between those who have and have not sought formal help specifically for premenstrual symptoms were examined using Mann-Whitney U tests and Chi-square tests, respectively. Predictive models of help-seeking were built using the decision tree-based machine learning method, Extreme Gradient Boosting (XGBoost). A total of 592 participants with complete data who endorsed premenstrual symptoms in consecutive cycles were included for analysis. Of those, 57.26% (n = 339) had previously seen a healthcare professional specifically for premenstrual symptoms. The model predicting formal help-seeking demonstrated fair performance, with an area under the receiver operating characteristic curve (AUROC) of 0.75 (SD = 0.06), a sensitivity of 0.65 (SD = 0.10), and a specificity of 0.71 (SD = 0.11). The strongest predictors of formal help-seeking were impaired social functioning, thinking that symptoms were severe, impairment in work/studies, and a previous poor care experience for gynaecological/reproductive conditions. These insights may be leveraged to encourage help-seeking behaviour, potentially reducing unnecessary distress or impairment. Improved knowledge of premenstrual symptoms and disorders is vital to facilitate identification of severe symptoms and impaired functioning related to the menstrual cycle. Additionally, improved guidance on when to seek help is required to increase the rate of formal help-seeking, particularly for individuals with high-risk psychological symptoms such as suicidality. More work is needed to determine the specific

**Data availability statement:** Due to the restrictions on data access specified in the ethical approval the study data cannot be shared outside of the research team. Researchers with specific queries about the data may contact the Cambridge Psychology Research Ethics Committee at the University of Cambridge (SBSEthics@admin.cam.ac.uk) who provided ethical approval for the current study.

**Funding:** This work was supported by the Stanley Medical Research Institute (https://stanleyresearch.org/; grant number 07R-1888). The funder had no role in study design, data collection and analysis, decision to publish, or preparation of the manuscript.

**Competing interests:** I have read the journal's policy, and the authors of this manuscript have the following competing interests: Prof. Sabine Bahn. is co-founder of and holds shares in Psyomics Ltd, is Director of Psynova Neurotech Ltd, and has a patent pending for dried blood spot biomarkers for bipolar disorder but declares no non-financial competing interests. Erin L. Funnell is a paid consultant for Psyomics Ltd but declares no non-financial competing interests. Jakub Tomasik received payments for licensing of research data and consulting not related to the current study from Psyomics Ltd and has a patent pending for dried blood spot biomarkers for bipolar disorder. The other authors declared that no competing interests exist.

mechanism by which previous poor care experiences drive further help-seeking for premenstrual symptoms.

## 1. Introduction

Premenstrual symptoms are a collection of psychological (e.g., low mood, irritability), behavioural (e.g., food cravings, overeating) and physical symptoms (e.g., headaches, bloating) occurring in the luteal phase of the menstrual cycle. Such symptoms are frequent in women and people assigned female at birth [1]. More severe presentations of premenstrual symptoms in consecutive cycles, with associated impaired functioning, can be indicative of premenstrual syndrome (PMS) or premenstrual dysphoric disorder (PMDD). The impact of these symptoms and disorders can be substantial [2,3], with evidence of an increased risk of suicidality, particularly among those suffering from PMDD [4–6].

There is evidence that despite the frequency of premenstrual symptoms and disorders [1,7,8] and their associated burdens, few seek formal help from a healthcare professional, with many instead turning to non-formal sources of help, such as the internet [9]. However, many available treatments for premenstrual symptoms and disorders require a prescription (e.g., oral contraceptives, antidepressants) or an onward referral to specialist healthcare services (e.g., talking therapies or gonadotropin releasing hormone (GnRH) analogue treatment). Therefore, not accessing formal healthcare services constitutes a potential barrier to receiving effective treatment for premenstrual symptoms and disorders.

As such, it is vital to identify factors which predict formal help-seeking for premenstrual symptoms and disorders. Identification of such factors may reveal avenues to encourage such health behaviours. Previous literature has identified some potential drivers of seeking treatment for premenstrual symptoms and disorders, with positive attitudes towards symptoms, older age, and higher symptom severity implicated [10]. However, given the changing relationship between the individual and healthcare providers in the digital age [11,12], it is important to re-evaluate the potential drivers of formal help-seeking. Machine learning offers a novel approach to examining the complex relationships between factors that may be involved in health behaviours, such as formal help-seeking [13]. Indeed, machine learning has already been applied to studying help-seeking for depressive symptoms [14] and post-partum depression [15]. Therefore, the aim of the current study was to utilise a machine learning approach to examine which symptoms, functional impairment factors, and barriers to accessing care may be predictive of formal help-seeking for premenstrual symptoms.

## 2. Materials and methods

### 2.1 Participants

Participants were recruited through both free posts on Facebook and Twitter, and paid advertisements on Facebook and Instagram between approximately the 10th of January 2024 and the 26th of February 2024. Inclusion criteria were: (1) ≥18 years,

(2) be assigned female at birth, (3) have a strong comprehension of the English language, (4) currently experiencing premenstrual symptoms, (5) not currently pregnant, in the perimenopause or post-menopausal, and (6) not diagnosed with any gynaecological conditions (e.g., endometriosis, polycystic ovary syndrome). Only participants who stated that they currently resided in the United Kingdom were included.

## 2.2. Materials and procedures

An online survey was delivered via the survey software Qualtrics XM®. The survey took between 10 and 20 minutes, and question flow was personalised based on previous answers given. The study materials were co-designed with an experienced consultant psychiatrist (SB). All participants were asked about sociodemographic characteristics, premenstrual symptoms and functional impairment, previous formal help-seeking behaviours, and perceived barriers to formal help-seeking as described below.

Premenstrual symptoms and associated functional impairment were measured using the Premenstrual Symptom Screening Tool (PSST) [16], a retrospective screening tool comprised of 19-items to assess premenstrual symptoms and associated functional impairment. In order to more closely align the PSST with the Diagnostic and Statistical Manual of Mental Disorders, Fifth Edition, Text Revision (DSM-5-TR) [17], we included a more specific timeframe of premenstrual symptoms to the question. As impaired romantic or intimate relationships are often associated with PMDD [17], this was added as an additional item to the PSST. Moreover, as increased suicidality is reportedly associated with PMS [7] and PMDD [5–7], an item to measure this was also added ("feeling like you don't want to be alive anymore or feeling suicidal"). Therefore, the total number of items of the PSST was increased from 19 to 21. All items were scored on a scale of "Not at all", "Mild", "Moderate", and "Severe". For the functional impairment items, the "Not at all" scale point was modified to "Not at all/not applicable". As a diagnosis of PMDD requires symptoms to be present in two consecutive cycles [17], a "yes"/"no" question was added to assess this.

Previous formal help-seeking specifically for premenstrual symptoms was assessed by asking the "yes"/"no" question: "Have you visited a healthcare professional specifically for your premenstrual symptoms?"

Barriers to formal help-seeking were evaluated using a version of the Barriers to Accessing Care Evaluation (BACE) scale [18] modified to be specific to help-seeking for premenstrual symptoms. The BACE is a scale assessing barriers to seeking care related to mental health concerns, and thus was deemed as relevant to the current study focus given that PMDD is included in the DSM [17]. The BACE includes 30 items scored on a scale from "Not at all" to "A lot", with a higher total score indicating more perceived barriers to care access. In the version of the BACE modified for this study, the questions displayed were adapted based on previous help-seeking experiences. If the participant reported never having seen a healthcare professional for their premenstrual symptoms they were asked: "Have any of these issues stopped you from getting professional care for your premenstrual symptoms?". If the participant had seen a healthcare professional for their premenstrual symptoms they were asked: "Did any of these issues initially delay or discourage you from getting, or continuing with, professional care for your premenstrual symptoms?" An additional 6 items were added to the BACE, expanding on existing items of the BACE (e.g., adding a secondary item to measure previous poor care experiences for gynaecological health conditions in addition to the existing item measuring previous poor care experiences for mental health conditions). These additional BACE items were based on previously identified potential barriers to formal help-seeking for premenstrual symptoms or premenstrual disorders [9,19,20].

## 2.3. Data analysis

Descriptive data analysis (i.e., means and standard deviations, frequencies and percentages) were analysed and processed in Excel, version 2206 (Microsoft Office 365). Group differences in ordinal and categorical data between those who have and have not sought formal help for premenstrual symptoms were examined using Mann-Whitney U tests and Chi-square tests, respectively. Tests of group differences were conducted in SPSS (version 29.0.1.1).

PLOS Mental Health

Predictive models of help-seeking were built using the decision tree-based machine learning method, Extreme Gradient Boosting (XGBoost) [21]. We employed a predictive modelling framework rather than an explanatory analysis as our goal was to explore the extent to which help-seeking behaviour could be predicted from available data, in addition to interpreting variable associations. Analysis was conducted in R version 4.4.0. Ordinal and numeric survey data were analysed as continuous variables, and categorical data (i.e., gender, ethnicity, and education) were encoded as dummy variables. Features that were duplicated, bijections, or constant were excluded from analysis. Age was excluded from the model as only data about age at the time of completing the survey and not age at the time of initial help-seeking was collected. The XGBoost algorithm was selected due to its capabilities in detecting non-linear relationships and interactions between variables, being robust to correlated and non-normally distributed features, and the fact that it does not require data transformations. It also offers explainability and interpretability, along with often superior performance compared to models such as logistic regression. The models were trained based on 10-fold stratified cross-validation repeated 10 times, with the number of folds and repeats selected for best replicability [22]. Following initial coarse tuning of a broader range of hyperparameters (10–1000 trees; learning rate 0.01 to 0.3; tree depth between 1 and 10; and alpha from 0 to 10), model parameters were fine-tuned for the number of trees (10–400), tree depth (1–3, to allow for interactions), and the learning rate (0.1 or 0.3), with the L1-regularisation parameter alpha fixed at 1. Model performance was evaluated using the out-of-sample cross-validated area under the receiver operating characteristic curve (AUROC). The optimal classification cut-off was determined using Youden's J statistic [23], to account for the imbalance between the groups. Reported values are averages and standard deviations obtained from the cross-validated models. Feature importance was assessed by measuring gain, i.e., the increase in accuracy brought by a feature to the branches it appeared on. The impact of features on model predictions was determined using the SHapley Additive exPlanations (SHAP) method [24]. Figures were prepared in the R package ggplot2.

### 2.4. Ethical approval

This study was approved by the University of Cambridge Psychology Research Ethics Committee (approval number PRE.2023.117). All participants provided written informed consent digitally prior to commencement of the survey.

## 3. Results

666 individuals consented to participate in the study. 88.88% (N = 592) with complete data (i.e., at least 97% completion) and endorsed premenstrual symptoms in consecutive cycles were included in the analysis.

### 3.1. Sociodemographic results

The mean age of respondents was 33.91 (Standard Deviation = 6.18, range = 18–51), with the majority identifying as women (97.30%, n = 576) and being white (91.89%, n = 544). For a full summary of sociodemographic characteristics, see S1 Data.

57.26% (n = 339) of the sample had sought help from a healthcare professional specifically for premenstrual symptoms.

The most frequently endorsed premenstrual symptom in the overall sample was increased feelings of anger/irritability (98.14%, n = 581; Table 1). The most frequently impaired domain of functioning was work/studies (94.26%, n = 558; Table 1). Mann-Whitney U tests indicated that there was a significant difference in the presence and severity of all symptoms and domains of functional impairment between help-seekers and non-help-seekers ($U_s > 30996.000$, $p \leq 0.005$; Table 1).

### 3.2. Predicting formal help-seeking

The model predicting help-seeking demonstrated fair performance (Fig 1) with an Area Under the Receiver Operating Characteristic Curve (AUROC) of 0.75 (SD = 0.06), a sensitivity of 0.65 (SD = 0.10), specificity of 0.71 (SD = 0.11), positive

**Table 1. Summary of premenstrual symptoms and functional impairment across the overall sample and by help-seeking status.**

| Symptom | Severity | Overall (N = 592) | | Previous help-seeker (n = 339) | | Non-help-seeker (n = 253) | | U | p |
|---|---|---|---|---|---|---|---|---|---|
| | | No. | % | No. | % | No. | % | | |
| Anger/ irritability | Not at all | 11 | 1.86 | 3 | 0.88 | 8 | 3.16 | 35923.000 | <.001 |
| | Mild | 107 | 18.07 | 49 | 14.45 | 58 | 22.92 | | |
| | Moderate | 303 | 51.18 | 174 | 51.33 | 129 | 50.99 | | |
| | Severe | 171 | 28.89 | 113 | 33.33 | 58 | 22.92 | | |
| Anxiety/ tension | Not at all | 20 | 3.38 | 7 | 2.06 | 13 | 5.14 | 32837.000 | <.001 |
| | Mild | 69 | 11.66 | 27 | 7.96 | 42 | 16.60 | | |
| | Moderate | 262 | 44.26 | 139 | 41.00 | 123 | 48.62 | | |
| | Severe | 241 | 40.71 | 166 | 48.97 | 75 | 29.64 | | |
| Tearfulness/ increased sensitivity to rejection | Not at all | 12 | 2.03 | 3 | 0.88 | 9 | 3.56 | 35487.000 | <.001 |
| | Mild | 76 | 12.84 | 39 | 11.50 | 37 | 14.62 | | |
| | Moderate | 241 | 40.71 | 123 | 36.28 | 118 | 46.64 | | |
| | Severe | 263 | 44.43 | 174 | 51.33 | 89 | 35.18 | | |
| Depressed mood/ hopelessness | Not at all | 27 | 4.56 | 8 | 2.36 | 19 | 7.51 | 37055.000 | .003 |
| | Mild | 107 | 18.07 | 57 | 16.81 | 50 | 19.76 | | |
| | Moderate | 232 | 39.19 | 130 | 38.35 | 102 | 40.32 | | |
| | Severe | 226 | 38.18 | 144 | 42.48 | 82 | 32.41 | | |
| Decreased interest in work activities | Not at all | 53 | 8.95 | 17 | 5.01 | 36 | 14.23 | 34326.000 | <.001 |
| | Mild | 141 | 23.82 | 74 | 21.83 | 67 | 26.48 | | |
| | Moderate | 246 | 41.55 | 144 | 42.48 | 102 | 40.32 | | |
| | Severe | 152 | 25.68 | 104 | 30.68 | 48 | 18.97 | | |
| Decreased interest in home activities | Not at all | 59 | 9.97 | 25 | 7.37 | 34 | 13.44 | 34224.000 | <.001 |
| | Mild | 150 | 25.34 | 74 | 21.83 | 76 | 30.04 | | |
| | Moderate | 242 | 40.88 | 140 | 41.30 | 102 | 40.32 | | |
| | Severe | 141 | 23.82 | 100 | 29.50 | 41 | 16.21 | | |
| Decreased interest in social activities | Not at all | 40 | 6.76 | 14 | 4.13 | 26 | 10.28 | 33758.500 | <.001 |
| | Mild | 120 | 20.27 | 51 | 15.04 | 69 | 27.27 | | |
| | Moderate | 249 | 42.06 | 153 | 45.13 | 96 | 37.94 | | |
| | Severe | 183 | 30.91 | 121 | 35.69 | 62 | 24.51 | | |
| Difficulty concentrating | Not at all | 42 | 7.09 | 11 | 3.24 | 31 | 12.25 | 30996.000 | <.001 |
| | Mild | 133 | 22.47 | 63 | 18.58 | 70 | 27.67 | | |
| | Moderate | 247 | 41.72 | 140 | 41.30 | 107 | 42.29 | | |
| | Severe | 170 | 28.72 | 125 | 36.87 | 45 | 17.79 | | |
| Fatigue/lack of energy | Not at all | 14 | 2.36 | 4 | 1.18 | 10 | 3.95 | 33440.500 | <.001 |
| | Mild | 62 | 10.47 | 24 | 7.08 | 38 | 15.02 | | |
| | Moderate | 230 | 38.85 | 120 | 35.40 | 110 | 43.48 | | |
| | Severe | 286 | 48.31 | 191 | 56.34 | 95 | 37.55 | | |
| Overeating/ food craving | Not at all | 33 | 5.57 | 18 | 5.31 | 15 | 5.93 | 37236.500 | .004 |
| | Mild | 148 | 25.00 | 74 | 21.83 | 74 | 29.25 | | |
| | Moderate | 250 | 42.23 | 139 | 41.00 | 111 | 43.87 | | |
| | Severe | 161 | 27.20 | 108 | 31.86 | 53 | 20.95 | | |
| Insomnia | Not at all | 194 | 32.77 | 89 | 26.25 | 105 | 41.50 | 35461.500 | <.001 |
| | Mild | 153 | 25.84 | 94 | 27.73 | 59 | 23.32 | | |
| | Moderate | 163 | 27.53 | 100 | 29.50 | 63 | 24.90 | | |
| | Severe | 82 | 13.85 | 56 | 16.52 | 26 | 10.28 | | |

*(Continued)*

**Table 1.** (Continued)

| Symptom | Severity | Overall (N = 592) | | Previous help-seeker (n = 339) | | Non-help-seeker (n = 253) | | | |
|---|---|---|---|---|---|---|---|---|---|
| | | No. | % | No. | % | No. | % | U | p |
| Needing more sleep | Not at all | 60 | 10.14 | 24 | 7.08 | 36 | 14.23 | 33111.000 | <.001 |
| | Mild | 134 | 22.64 | 60 | 17.70 | 74 | 29.25 | | |
| | Moderate | 237 | 40.03 | 144 | 42.48 | 93 | 36.76 | | |
| | Severe | 161 | 27.20 | 111 | 32.74 | 50 | 19.76 | | |
| Feeling overwhelmed or out of control | Not at all | 29 | 4.90 | 8 | 2.36 | 21 | 8.30 | 33666.000 | <.001 |
| | Mild | 91 | 15.37 | 45 | 13.27 | 46 | 18.18 | | |
| | Moderate | 223 | 37.67 | 117 | 34.51 | 106 | 41.90 | | |
| | Severe | 249 | 42.06 | 169 | 49.85 | 80 | 31.62 | | |
| Physical symptoms | Not at all | 16 | 2.70 | 7 | 2.06 | 9 | 3.56 | 35999.000 | <.001 |
| | Mild | 119 | 20.10 | 56 | 16.52 | 63 | 24.90 | | |
| | Moderate | 269 | 45.44 | 151 | 44.54 | 118 | 46.64 | | |
| | Severe | 188 | 31.76 | 125 | 36.87 | 63 | 24.90 | | |
| Feeling suicidal* | Not at all | 232 | 39.19 | 118 | 34.81 | 114 | 45.06 | 37370.500 | .005 |
| | Mild | 154 | 26.01 | 92 | 27.14 | 62 | 24.51 | | |
| | Moderate | 130 | 21.96 | 75 | 22.12 | 55 | 21.74 | | |
| | Severe | 76 | 12.84 | 54 | 15.93 | 22 | 8.70 | | |
| **Impairment** | **Severity** | n | % | n | % | n | % | U | p |
| Work/studies | Not at all/NA | 34 | 5.74 | 9 | 2.65 | 25 | 9.88 | 28701.000 | <.001 |
| | Mild | 167 | 28.21 | 72 | 21.24 | 95 | 37.55 | | |
| | Moderate | 269 | 45.44 | 159 | 46.90 | 110 | 43.48 | | |
| | Severe | 122 | 20.61 | 99 | 29.20 | 23 | 9.09 | | |
| Relationship with co-workers | Not at all/N.A. | 182 | 30.74 | 75 | 22.12 | 107 | 42.29 | 30340.00 | <.001 |
| | Mild | 211 | 35.64 | 119 | 35.10 | 92 | 36.36 | | |
| | Moderate | 157 | 26.52 | 110 | 32.45 | 47 | 18.58 | | |
| | Severe | 42 | 7.09 | 35 | 10.32 | 7 | 2.77 | | |
| Romantic or intimate relationships* | Not at all/N.A. | 64 | 10.81 | 31 | 9.14 | 33 | 13.04 | 32928.500 | <.001 |
| | Mild | 134 | 22.64 | 60 | 17.70 | 74 | 29.25 | | |
| | Moderate | 214 | 36.15 | 117 | 34.51 | 97 | 38.34 | | |
| | Severe | 180 | 30.41 | 131 | 38.64 | 49 | 19.37 | | |
| Family | Not at all/N.A. | 83 | 14.02 | 33 | 9.73 | 50 | 19.76 | 34395.000 | <.001 |
| | Mild | 181 | 30.57 | 99 | 29.20 | 82 | 32.41 | | |
| | Moderate | 235 | 39.70 | 138 | 40.71 | 97 | 38.34 | | |
| | Severe | 93 | 15.71 | 69 | 20.35 | 24 | 9.49 | | |
| Social life | Not at all/N.A. | 65 | 10.98 | 15 | 4.42 | 50 | 19.76 | 30115.000 | <.001 |
| | Mild | 168 | 28.38 | 81 | 23.89 | 87 | 34.39 | | |
| | Moderate | 239 | 40.37 | 160 | 47.20 | 79 | 31.23 | | |
| | Severe | 120 | 20.27 | 83 | 24.48 | 37 | 14.62 | | |
| Home responsibilities | Not at all/N.A. | 48 | 8.11 | 17 | 5.01 | 31 | 12.25 | 29562.000 | <.001 |
| | Mild | 179 | 30.24 | 81 | 23.89 | 98 | 38.74 | | |
| | Moderate | 242 | 40.88 | 141 | 41.59 | 101 | 39.92 | | |
| | Severe | 123 | 20.78 | 100 | 29.50 | 23 | 9.09 | | |

*Key.* *: Items were researcher-generated for the current study and presented alongside the PSST; N.A.: Not applicable.

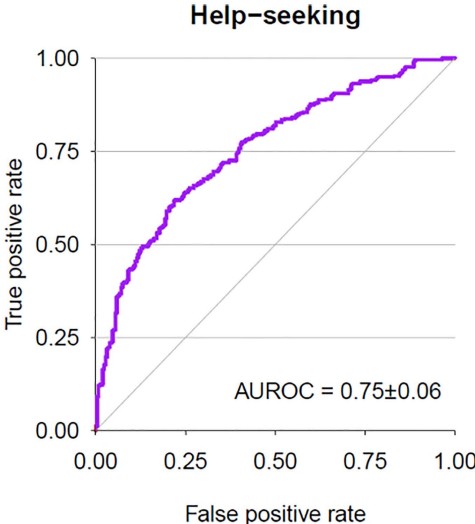

**Fig 1. Model performance for prediction of formal help-seeking specifically for premenstrual symptoms. Key. AUROC: Area Under the Receiver Operating Characteristic Curve.**

predictive value of 0.76 (SD = 0.07), and negative predictive value of 0.61 (SD = 0.07). Median number of participants across the training sets was 533, and 59 across the test sets.

The strongest predictor of formal help-seeking was impaired social functioning, with higher social impairment associated with a higher likelihood of formal help-seeking for premenstrual symptoms (Fig 2). Other influential predictors of formal help-seeking were thinking that symptoms were severe enough, higher impairment in work/studies, and a previous poor care experience for gynaecological/reproductive conditions.

## 4. Discussion

The current study set out to identify which symptoms, domains of functional impairment, and barriers to accessing care may be predictive of formal help-seeking for premenstrual symptoms by utilising machine learning methodology. Data demonstrates that those who have previously sought help endorse a higher frequency and severity of symptoms and functional impairment. Notably, a high proportion of the overall sample endorsed experiencing suicidality, with a substantial number of non-help-seekers reporting mild, moderate or severe suicidality. The machine learning approach revealed individual factors implicated in prediction of formal help-seeking, with model performance having a fair predictive power as indicated by an AUROC of 0.75, sensitivity of 0.65, and a specificity of 0.71.

Impaired social functioning was identified as a core predictor of formal help-seeking for premenstrual symptoms from the data explored in the current study. Previous research has also identified that diminished social functioning is highly predictive of women contacting a general practitioner to discuss mental health symptoms [25]. Evidence indicates that social functioning is more impaired in individuals with PMDD than healthy controls across various phases of the menstrual cycle, with improving social connectedness suggested as a candidate for intervention [26]. In addition to social impairment, impairment in work or studies was also identified as a predictor of formal help-seeking. This domain of impairment is common in individuals with premenstrual symptoms and disorders, with more severe symptoms being linked to higher levels of both presenteeism and absenteeism [27,28]. There are also wider impacts across other phases of the menstrual cycle, with qualitative reports showing the use of compensatory behaviours to account for reduced performance

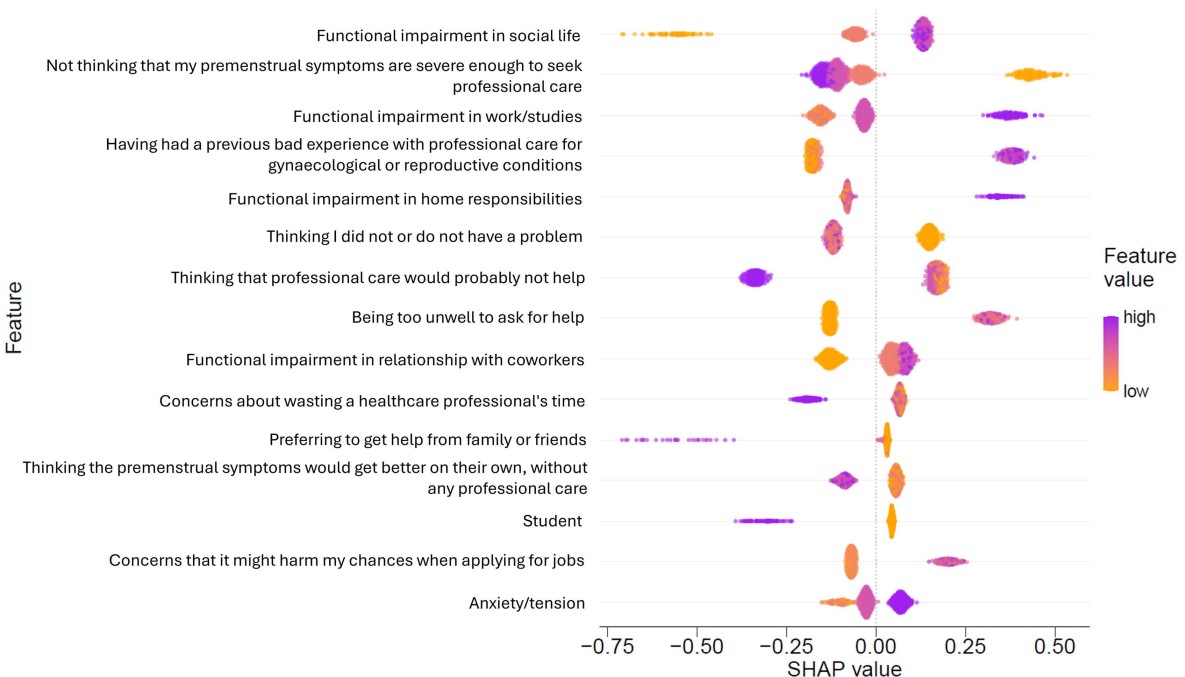

**Fig 2. SHAP values for the top 15 factors implicated in the model.** Features in the figure are ordered by decreasing feature importance. SHAP values below 0 show lower likelihood of formal help-seeking and SHAP values above 0 show higher likelihood of formal help-seeking.

during the luteal period [29]. It is worth noting that much of the perceived impairments in the workplace may be driven by social impairments, with reports of increased irritability and conflict [29]. Improving identification of impaired functioning during the luteal phase of the menstrual cycle may be important in encouraging help-seeking behaviours. One approach to achieve this is the use of period tracking tools, such as mobile applications, which have shown efficacy in enhancing awareness of the menstrual cycle's impact on mental health and behaviour [30], with early evidence indicating that such apps may promote formal help-seeking [31]. An additional approach is psychoeducation, which has shown efficacy at improving functioning for mental health conditions such as bipolar disorder [32], as well as increasing help-seeking for depressive symptoms [33,34]. Psychoeducation for premenstrual symptoms and disorders has also demonstrated efficacy for symptom reduction [35,36], including when delivered digitally [37]. There is interest in digital psychoeducation for mental health symptoms related to the menstrual cycle [38], but further work is required to comprehensively determine the possible benefits of psychoeducation for premenstrual symptoms and disorder in addition to symptom reduction, including its potential ability to encourage formal help-seeking behaviour.

Thinking that symptoms were severe enough was associated with a higher likelihood of formal help-seeking. This is in line with previous research [10], also implying that an individual must first identify the presence of problematic symptoms in order to seek help. This finding is concerning, as around 55% of those who had *not* previously sought formal help for premenstrual symptoms endorsed suicidality prior to their period. This indicates that a substantial proportion of non-formal help-seekers are experiencing high-risk, and likely highly distressing, psychological symptoms which are unmonitored and remain untreated.

This result reveals a vital need to improve public health awareness around premenstrual symptoms and disorders, particularly in regard to symptom recognition and guidance on when it may be appropriate to seek help. Interventions intended to improve mental health literacy demonstrate short-term increases in formal help-seeking behaviour, however

there is a need to evaluate their long-term efficacy [39]. Further, it is important to ensure interventions are specific to the population of interest to guarantee their relevancy [39]. For example, it is important to recognise that perceived symptom severity of premenstrual symptoms will likely be more complex to conceptualise compared to other disorders due to the clear trigger and cyclical nature with symptoms intermittently improving for many individuals without any intervention. Therefore, more research is required to understand how perceptions of premenstrual symptom severity are formed and held. Such understanding will be important in the design of effective public health education campaigns intended to improve recognition of severe and potentially high-risk symptoms and encourage formal help-seeking where appropriate. It will also be important to identify the most appropriate settings and modalities for the delivery of public health education. Schools are a commonly utilised setting for the delivery of interventions intending to improve help-seeking behaviour for mental health [40]. Health education interventions delivered in school settings have also demonstrated efficacy in the reduction of premenstrual symptom severity [41,42] and increasing mental health literacy [43]. However, further research is needed in larger and more diverse samples, with long-term follow up of outcomes also required to determine sustained benefits [40]. Additionally, as evidence demonstrates public health interventions delivered in schools are rarely sustainable [44], it is vital to co-design these interventions with key stakeholders to ensure both initial uptake and longer-term implementation. Additionally, given the trajectory of premenstrual symptoms and disorders is highly variable [45], it will be important that high-quality informational resources are available across the reproductive lifespan.

Aside from public health education, the potential lack of recognition of severe or high-risk symptoms indicates a potential need for accessible and comprehensive symptom screening which offers signposting recommendations. Such screening and signposting tools could reduce uncertainty around whether symptoms are severe enough to warrant formal help-seeking.

Perhaps surprisingly, previous *poor* care experiences for gynaecological or reproductive conditions were associated with a higher likelihood of formal help-seeking. Previous help-seeking, irrespective of the perceived quality of the care interaction, has previously been identified as a predictor of formal help-seeking for mental health symptoms [46]. However, contrary to our findings, we would expect poor previous care experiences to have *reduced* likelihood of formal help-seeking, with previous evidence demonstrating previous positive experience was associated with facilitating healthcare access [40]. Notably, qualitative evidence has demonstrated recurring help-seeking for premenstrual disorders with reported repeated dismissals by healthcare professionals [19]. In this regard, affected individuals may request follow-up appointments to advocate for or demand better care. This may include seeking a specific treatment or seeking help from a different healthcare professional. This need for repeated advocacy to receive care may explain why previous poor care experiences may drive further formal help-seeking for premenstrual symptoms. Additionally, formal help seeking for those with poor previous care experience could be due to ineffective formal management which results in remaining symptomatic. More work is required to disentangle the influence of previous help-seeking experiences, particularly when negative, on future formal help-seeking for premenstrual symptoms.

There are several limitations in this study which need to be considered alongside the results. Firstly, the participant demographics have likely resulted in limited generalisability, with the majority being white, identifying as women, and being well-educated. Therefore, the findings may not be applicable to individuals from other ethnic groups or gender-diverse backgrounds. Moreover, due to the recruitment methods employed in the current study, the sample may not be representative of the broader population, potentially limiting the generalisability of the findings outside of those who frequently engage in online spaces such as social media. As such, further research is required to validate this model in a larger sample with a more diverse study population utilising varied recruitment strategies including non-digital approaches. Such work will be important for the development of help-seeking interventions that are relevant and effective across diverse demographic groups.

Secondly, given the multifactorial nature of help-seeking, there are other predictors which are likely to be highly predictive but were not assessed in the current study. Future studies may wish to incorporate further factors previously implicated in help-seeking, such as social support [47] and symptom knowledge [48]. Moreover, future work would benefit from

including a wider range of sociodemographic factors in predictive modelling such as socioeconomic status or deprivation indicators which are associated with increased practical barriers to accessing care [49]. Beyond these specific variables, future research should aim to co-design of the study materials with individuals who have lived experience of premenstrual symptoms. Doing so would help ensure that the model includes a more comprehensive set of factors that individuals from the population of interest consider important when deciding whether to seek formal care. Additionally, collecting qualitative data on help-seeking facilitators would likely offer a more nuanced perspective and deliver further insights informed by lived experience.

Finally, the current study may have inadvertently excluded some previous help-seekers by narrowly defining help-seeking status as healthcare sought *specifically for* premenstrual symptoms. Given that the literature shows multiple health complaints are often raised in a single appointment with healthcare professionals [50,51], it is possible that some individuals labelled as non-previous help-seekers in the current study may have discussed their premenstrual symptoms with a healthcare professional during the course of help-seeking for another complaint.

## 5. Conclusions

To conclude, the current study identified predictors of formal help-seeking for premenstrual symptoms, pointing to the need for improved education to improve help-seeking behaviours. More work is needed to fully realise the potential benefits of tracking and psychoeducation, delivered digitally or non-digitally, for improving identification of symptoms and associated functional impairment. Long-term research will be important for assessing the ability of these tools and resources to encourage formal help-seeking. Crucially, the results also reveal a need to improve public health education, ideally early in life, to facilitate the early identification of severe and high-risk symptoms. The study revealed a potentially substantial proportion of individuals who are experiencing premenstrual suicidality but are not known to healthcare professionals. Therefore, it will also be vital for guidance on when help-seeking is appropriate to remove the uncertainty for individuals experiencing premenstrual symptoms. Further work is required to disentangle the influence of previous help-seeking on future and continued help-seeking for premenstrual symptoms, with it being important to determine the motivation for repeated help-seeking and how to ensure needs are effectively met in the initial contact.

## Supporting information

**S1 Data. Summary of sociodemographic characteristics.**
(DOCX)

## Author contributions

**Conceptualization:** Erin Lucy Funnell, Nayra A. Martin-Key, Sabine Bahn.

**Formal analysis:** Jakub Tomasik.

**Funding acquisition:** Sabine Bahn.

**Investigation:** Erin Lucy Funnell, Nayra A. Martin-Key, Jakub Tomasik.

**Methodology:** Erin Lucy Funnell, Nayra A. Martin-Key, Jakub Tomasik.

**Supervision:** Sabine Bahn.

**Visualization:** Jakub Tomasik.

**Writing – original draft:** Erin Lucy Funnell.

**Writing – review & editing:** Erin Lucy Funnell, Nayra A. Martin-Key, Jakub Tomasik, Sabine Bahn.

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
