## [Decision Letter · Decision Letter 0]

8 Apr 2025

PMEN-D-25-00045

Identifying Predictors of Formal Help-Seeking for Premenstrual Symptoms: a Machine Learning Analysis of Symptom, Functional Impairment and Barriers Data

PLOS Mental Health

Dear Dr. Funnell,

Thank you for submitting your manuscript to PLOS Mental Health. After careful consideration, we feel that it has merit but does not fully meet PLOS Mental Health’s publication criteria as it currently stands. Therefore, we invite you to submit a revised version of the manuscript that addresses the points raised during the review process.

Authors should consider the different concerns and issues raised by reviewers.

We look forward to receiving your revised manuscript.

Kind regards,

Ariel Soares Teles

Academic Editor

PLOS Mental Health

Journal Requirements:

1. Please note that PLOS Mental Health has specific guidelines on code sharing for submissions in which author-generated code underpins the findings in the manuscript. In these cases, we expect all author-generated code to be made available without restrictions upon publication of the work. Please review our guidelines at https://journals.plos.org/mentalhealth/s/materials-and-software-sharing#loc-sharing-code and ensure that your code is shared in a way that follows best practice and facilitates reproducibility and reuse.

Additional Editor Comments (if provided):

Reviewers' comments:

Reviewer's Responses to Questions

**Comments to the Author**

1. Does this manuscript meet PLOS Mental Health’s publication criteria?

Reviewer #1: Yes

Reviewer #2: Yes

2. Has the statistical analysis been performed appropriately and rigorously?

Reviewer #1: Yes

Reviewer #2: No

3. Have the authors made all data underlying the findings in their manuscript fully available (please refer to the Data Availability Statement at the start of the manuscript PDF file)?

Reviewer #1: Yes

Reviewer #2: Yes

4. Is the manuscript presented in an intelligible fashion and written in standard English?

Reviewer #1: Yes

Reviewer #2: Yes

Reviewer #1: This manuscript was well written and the machine learning strategy is certainly a technique matching the objectives set out by the authors. The conclusions followed from the analyses in that the current study identified predictors of formal help-seeking for premenstrual symptoms, possibly pointing to the need for improved education to improve help-seeking behaviors. The limitations were duly noted by the investigators. However, despite the limitations, the information was processed and presented successfully.

The sample size was certainly adequate. The XGBoost algorithm was properly selected due to its capabilities in detecting non-linear relationships and interactions between variables, being robust to correlated and non-normally distributed features. Model performance and validation were conducted. The AOC was reasonable. It would have been helpful to have a bit more detail such as size of training and test sets and the performance of the confusion matrix in the process, unless this reviewer missed something.

Statistically, the prediction performance was well explained and the comparison of the help-seekers and non-help-seekers using the Mann-Whitney U tests clearly indicated (Table1) that there was a significant difference in the presence and severity of all symptoms and domains of functional impairment.

Reviewer #2: This paper addresses an important and highly relevant topic related to women's wellbeing, contributing to the de-stigmatization of conversations around menstrual symptoms and investigating the barriers and predictors of help-seeking behavior.

The paper is well-written, with a particularly strong introduction and discussion of both the benefits and barriers to seeking help. The sample size is relatively large (592 participants in total, 339 of whom sought help), covering a broad age range (18–55).

The methodology appears to be a weaker aspect of the study:

Firstly, while XGBoost is a powerful algorithm, research suggests that logistic regression often performs just as well for tabular health data while being easily interpretable. Given the number of predictors, Lasso Logistic Regression (LR) would be a natural first choice, with XGBoost serving as a supplementary approach to explore interaction effects. For these reasons, I strongly advise the authors to run and validate Lasso LR along with XGBoost.

Secondly, hyperparameter tuning should be reconsidered. The current XGBoost hyperparameter space is dangerously narrow (number of trees: 1–100, learning rates: 2 options, tree depth: 1–2), and expanding it may improve algorithms’ performance and give more robust results. A more suitable range may include 50–1000 trees, learning rate of 0.01–0.3 and tree depth between 2 and 10. In addition, including L1-regularisation (alpha parameter 0 - 10) is recommended given a large number of predictor compared to the sample size.

Thirdly, some clarification on Validation Metrics and predictive (ML) approach justification are needed. It appears that AUC and accuracy are reported from the validation metrics, specifically AUC from out-of-sample participants across 10 10-fold cross-validations - could the authors explicitly clarify this in the text. Further, the decision to take a predictive approach should be better justified. An alternative would be to use multinomial regression or Lasso Logistic Regression to analyze the impact of various exposures on the likelihood of seeking help. In my opinion, both approaches are valid, but the strengths of the chosen predictive methodology should be clearly articulated.

Finally, representativeness of the sample remains a key concern. I appreciate that this limitation is acknowledged, but I would also note that the sample may be biased toward individuals active on social media, and can be further skewed by the platforms’ algorithms in determining whom to show the ad. Additionally, the country of participants is not reported, which is significant given that healthcare systems and access barriers vary widely. For instance, anecdotal evidence from the UK suggests that seeking professional help often requires framing symptoms in terms of their impact on work and daily responsibilities, as personal pain and discomfort related to menstruation are more easily dismissed.

To further strengthen the study and future research in this area, I suggest mentioning the importance of inclusion of the socioeconomic indicators along with professional status (even a crude measure of participants’ wealth or socioeconomic status could provide more context for help-seeking behaviors) and engaging PPI (Patient and Public Involvement) Groups in study conceptualisation which could enhance the design and help identify previously missed symptoms, barriers, and their perceived importance.

In conclusion, this study is well-presented, with a carefully designed and well-described methodology and questionnaires choice and modifications. The topic is socially important, and I encourage improvements in the data analysis to strengthen its potential for publication. Moreover, despite the above-mentioned concerns, given a relatively large sample size and the high prevalence of distressing symptoms such as suicidal thoughts, it would be great to see this study being published once methodological concerns are addressed.

**Do you want your identity to be public for this peer review?** For information about this choice, including consent withdrawal, please see our Privacy Policy

Reviewer #1: No

Reviewer #2: **Yes: ** Diana Shamsutdinova, PhD

---

## [Decision Letter · Decision Letter 1]

2 Jul 2025

Identifying Predictors of Formal Help-Seeking for Premenstrual Symptoms: a Machine Learning Analysis of Symptom, Functional Impairment and Barriers Data

PMEN-D-25-00045R1

Dear Miss Funnell,

We are pleased to inform you that your manuscript 'Identifying Predictors of Formal Help-Seeking for Premenstrual Symptoms: a Machine Learning Analysis of Symptom, Functional Impairment and Barriers Data' has been provisionally accepted for publication in PLOS Mental Health.

Best regards,

Ariel Soares Teles

Academic Editor

PLOS Mental Health

Reviewer Comments (if any, and for reference):

Reviewer's Responses to Questions

**Comments to the Author**

Reviewer #1: All comments have been addressed

Reviewer #2: All comments have been addressed

publication criteria?

Reviewer #1: Yes

Reviewer #2: Yes

3. Has the statistical analysis been performed appropriately and rigorously?

Reviewer #1: Yes

Reviewer #2: Yes

4. Have the authors made all data underlying the findings in their manuscript fully available (please refer to the Data Availability Statement at the start of the manuscript PDF file)?

Reviewer #1: Yes

Reviewer #2: Yes

5. Is the manuscript presented in an intelligible fashion and written in standard English?

Reviewer #1: Yes

Reviewer #2: Yes

Reviewer #1: (No Response)

Reviewer #2: Thank you for addressing the concerns mentioned in the initial review and making appropriate changes in the manuscript. I do not have any further comments.

**Do you want your identity to be public for this peer review?** For information about this choice, including consent withdrawal, please see our Privacy Policy

Reviewer #1: No

Reviewer #2: No
